# WEAR:
# AN OUTDOOR SPORTS DATASET FOR WEARABLE AND EGOCENTRIC ACTIVITY RECOGNITION

## ABSTRACT

Though research has shown the complementarity of camera- and inertial-based data, datasets which offer both egocentric video and inertial-based sensor data remain scarce. In this paper, we introduce WEAR, an outdoor sports dataset for both vision- and inertial-based human activity recognition (HAR). The dataset comprises data from 18 participants performing a total of 18 different workout activities with untrimmed inertial (acceleration) and camera (egocentric video) data recorded at 10 different outside locations. Unlike previous egocentric datasets, WEAR provides a challenging prediction scenario marked by purposely introduced activity variations as well as an overall small information overlap across modalities. Benchmark results obtained using each modality separately show that each modality interestingly offers complementary strengths and weaknesses in their prediction performance. Further, in light of the recent success of temporal action localization models following the architecture design of the Action-Former, we demonstrate their versatility by applying them in a plain fashion using vision, inertial and combined (vision + inertial) features as input. Results demonstrate both the applicability of vision-based temporal action localization models for inertial data and fusing both modalities by means of simple concatenation, with the combined approach (vision + inertial features) being able to produce the highest mean average precision and close-to-best F1-score. The dataset and code to reproduce experiments is publicly available via: https://www.anonymous.edu/anon

## 1 INTRODUCTION

The physical activities that we perform in our daily lives have been identified as valuable information for a number of research fields and applications, such as work processes support, preventive healthcare, cognitive science or workout monitoring (e.g., Bao & Intille (2004); Patterson et al. (2005); Ward et al. (2006)). Research efforts have till now shown that physical activities can be detected using either wearable inertial sensors or camera-based approaches. The inertial sensors can continuously observe motion and gestures at particular body locations, whereas camera-based systems can typically observe the user's entire body, but can be hindered by (self-)occlusions. Inertial data manifests itself as multidimensional timeseries, while image data can be interpreted more easily afterwards. Even though research has shown (e.g. Spriggs et al. (2009); Song et al. (2016a); Diete et al. (2019); Nakamura et al. (2017)) that both modalities are complementary to each other, available benchmark datasets that provide both egocentric video and inertial-based sensor data remain scarce. We therefore introduce WEAR, an outdoor sports human activity recognition (HAR) dataset featuring workout activities performed by 18 participants while wearing inertial sensors on both wrists and ankles as well as a head-mounted camera capturing egocentric vision using a wide field-of-view - see Figure 1. In light with one of the key challenges in HAR, namely the *NULL*-class problem (Bulling et al., 2014), WEAR provides continuous data streams of each workout session including all breaks and interruptions. Our dataset features a challenging prediction scenario marked by purposely introduced activity variations, activities consisting of within-activity sequences (i.e. a sequence of multiple base activities) and an overall small information overlap across modalities. Unlike previous egocentric datasets, included activities are not defined by human-object-interaction

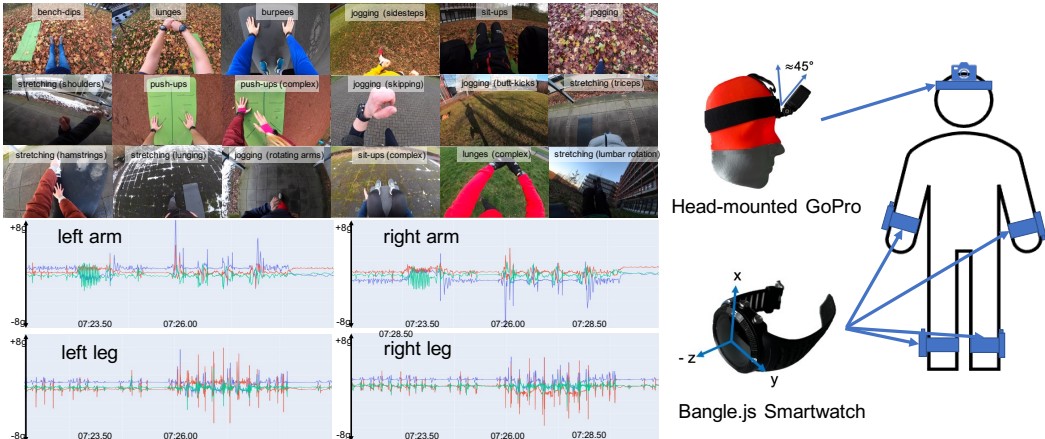

Figure 1: Setup and example data from the two types of wearable sensors in our dataset. Participants were equipped with four open-source smartwatches (one per limb) and a head-mounted camera.

(e.g. DelPreto et al. (2022); de la Torre et al. (2009)) nor originate from inherently distinct activity categories (e.g. Possas et al. (2018); Xu et al. (2023)). WEAR was collected at 10 different outdoor recording locations, with each location introducing different visual and surface conditions, yet not providing cues about the activity being performed. With these dataset traits in place we deem the WEAR dataset being an exemplary dataset to assess methods on how to combine both inertial- and vision-based features in the context of HAR. Our contributions in this paper are three-fold:

1. We introduce a new inertial- and vision-based HAR dataset called WEAR. The dataset features data of 18 participants, each performing 18 different sports activities.

2. We provide benchmark scores using both wearable- (Bock et al., 2021; Abedin et al., 2021) and vision-based (Zhang et al., 2022; Shi et al., 2023) state-of-the art models.

3. We demonstrate that state-of-the-art temporal action localization models from computer vision are excellently suited to not only process raw inertial data, but even successfully fuse multi-modal information significantly outperforming the best single-modality approach as well as beating the best possible (oracle) late fusion approach in terms of mAP.

## 2 RELATED WORK

**Inertial-based HAR** Compared to video-based modalities body-worn sensor systems bear a great potential in analyzing our daily activities with minimal intrusion, yielding various applications from the provision of medical support to supporting complex work processes (Bulling et al., 2014). Within the last decade deep learning based-methods have established themselves as the de facto standard in inertial-based HAR as they have shown to outperform classical machine learning algorithms (Ordóñez & Roggen, 2016; Hammerla et al., 2016; Guan & Plötz, 2017). One of the most well-known deep learning approaches for inertial-based HAR is the *DeepConvLSTM* which is a hybrid model combining both convolutional and recurrent layers (Ordóñez & Roggen, 2016). By combining both types of layers the network is able to automatically extract discriminative features and model temporal dependencies. Following the success of the original DeepConvLSTM, researchers worked on extending the architecture (Murahari & Plötz, 2018; Xi et al., 2018) or build up on the idea of combining convolutional and recurrent layers by proposing their own architectures (Xu et al., 2019; Abedin et al., 2021; Yuki et al., 2018; Zhou et al., 2022). Within this publication we are reporting benchmark scores using the WEAR dataset inertial sensor-streams as input for two popular HAR models (Bock et al., 2021; Abedin et al., 2021). Contrary to the belief that one needs to employ multiple recurrent layers when dealing with sequential data (Karpathy et al., 2015), Bock et al. (2021) proposed an altered *shallow DeepConvLSTM* architecture which proved to outperform the original architecture by a significant margin. Differently, Abedin et al. (2021) chose to build up on the idea

of the DeepConvLSTM and introduced the *Attend-and-Discriminate* architecture which exploits interactions among different sensor modalities by introducing self-attention through a cross-channel interaction encoder and adding attention to the recurrent parts of the network.

**Vision-based HAR** Predicting activities performed by humans based on visual-cues can broadly be categorized into three main application scenarios: action recognition, localization and anticipation. Action recognition systems (Liu et al., 2021b; Wang et al., 2021; Li et al., 2022) aim to assign a set of trimmed action segments an activity label. Contrarily, temporal action localization systems (Zhang et al., 2022; Yang et al., 2022; Liu et al., 2022b) are tasked to identify start and end times of all activities in a untrimmed video by predicting a set of activity triplets *(start, end, activity label)*. Lastly, action anticipation systems (Girdhar & Grauman, 2021; Roy & Fernando, 2022) aim to predict the label of a future activity having observed a segment preceding its occurrence. Though sensor-based HAR systems are employed using a sliding window approach and thus assign activity labels to a set of trimmed inertial-sequences, their ultimate goal is to identify a set of activities within a continuous timeline. We therefore deem vision-based temporal action localization to be most comparable to inertial-based HAR and will focus on it in our benchmark analysis. Existing temporal action localization methods can be divided into two categories: two- and single-stage approaches. Two-stage approaches (Lin et al., 2019; 2020; Xu et al., 2020; Bai et al., 2020; Zhao et al., 2020; Zeng et al., 2019; Gong et al., 2020; Liu et al., 2021a; Qing et al., 2021; Sridhar et al., 2021; Zhu et al., 2021; Zhao et al., 2021; Tan et al., 2021) divide the process of temporal action localization into two subtasks. First, during the action segment proposal generation, candidate video segments are generated which are then, classified with an activity label as well as refined regarding their temporal boundaries. Contrarily, single-stage approaches (Yang et al., 2022; Shi et al., 2022; Nag et al., 2022; Liu et al., 2022b; Liu & Wang, 2020; Long et al., 2019; Lin et al., 2021; Chen et al., 2022; Zhang et al., 2022; Shi et al., 2023) aim to localize actions in a single shot without using action proposals.

In light with the success of transformer architectures in natural language processing (see e.g. Vaswani et al. (2017); Devlin et al. (2019)) and computer vision (see e.g. Kolesnikov et al. (2021); Yuan et al. (2021); Liu et al. (2021b)), researchers have demonstrated their applicability for temporal action localization (Cheng & Bertasius, 2022; Liu et al., 2022a;b; Shi et al., 2022; Tan et al., 2021; Zhang et al., 2022) breaking previously held benchmark scores of numerous popular datasets (Heilbron et al., 2015; Damen et al., 2022; Jiang et al., 2014) without any additional training data by a significant margin. One of such architectures is the *ActionFormer* proposed by Zhang et al. (2022), which is an end-to-end trainable transformer-based architecture, which unlike other single-stage approaches, does not rely on pre-defined anchor windows. The architecture combines multiscale feature representations with local self-attention and is trained through a classification and regression loss calculated by a light-weighted decoder. Building up on the works of Zhang et al. (2022), Shi et al. (2023) proposed the *TriDet* model which suggest to replace the transformer layers of the ActionFormer with fully-convolutional, so-called SGP layers, as well as use a trident regression head which claims to improve imprecise boundary predictions via an estimated relative probability distribution around the boundary. Given the rapid rise in popularity of single-stage temporal action localization such as the ActionFormer, we decided said models to be a suited option to deliver a first benchmark for the WEAR dataset.

**Multimodal (Inertial and RGB Video) HAR** In Table 1 we show a curated list of datasets which provide both egocentric vision- (e.g. RGB, depth) and IMU-based (e.g. accelerometer, gyroscope, magnetometer) modalities in the context of HAR. We compare datasets regarding their recency, number of participants, number and type of activities performed, recording environment, camera and IMU position and whether the dataset is provided on a clip-basis or a continuous stream. As evident by the rise in popularity of commercial head-mounted cameras and wrist-worn smartwatches for tracking sports, we decided to position the camera and IMU sensors used during collection of the WEAR dataset in line with the recent trends in real-world application scenarios. With the head and limbs being positions which do not limit participants in their freedom of movement, we deem said positions to further be most suited in capturing how participants interact with their environment and/ or objects. This makes the works of de la Torre et al. (2009), Song et al. (2015) Diete et al. (2019) and DelPreto et al. (2022) to be most comparable to the WEAR dataset. DelPreto et al. (2022) and de la Torre et al. (2009) both provide datasets of participants cooking food recipes. Different from the WEAR dataset, recording takes places indoors in an artificial kitchen environment, which by nature

Table 1: List of available egocentric vision datasets, which provide inertial data, compared with the WEAR dataset. We differentiate between recency (year), number and type of activity classes (S = Sports, G = Gestures, L = Locomotion, D = Daily Living, C = Cooking, O = Other), number of subjects, recording environment (laboratory, outside or inside), location of the Camera and IMU sensor (Multi = multiple locations on body) and recording type (trimmed or untrimmed video sequences).

| Dataset | General | | | | | Sensor Location | | |
|---|---|---|---|---|---|---|---|---|
| | Year | Sbjs | Cls | Type | Where | Camera | IMU | Recording |
| CMU-MMAC de la Torre et al. (2009) | 2009 | 16 | 29 | C | Lab | Head | Multi | Untrimmed |
| MEAD Song et al. (2015) | 2015 | 2 | 20 | A | In/Out | Head | Head | Trimmed |
| Stanford-ECM Nakamura et al. (2017) | 2017 | 10 | 24 | S | In/Out | Chest | Chest | Trimmed |
| Daily Intention Wu et al. (2017) | 2017 | 12 | 34 | D | In | Wrist | Arms | Trimmed |
| DataEgo Possas et al. (2018) | 2018 | 8-10 | 20 | D | In/Out | Head | Head | Trimmed |
| ADL Dataset Diete et al. (2019) | 2019 | 2 | 6 | D | In | Head | Wrists | Untrimmed |
| Ego4D Grauman et al. (2022) | 2021 | 931 | 110 | D | In/Out | Head | Head | Untrimmed |
| ActionSense DelPreto et al. (2022) | 2022 | 10 | 20 | C | Lab | Head | Multi | Untrimmed |
| EPIC-Kitchens Damen et al. (2022) | 2022 | 37 | ≈149 | C | In | Head | Head | Untrimmed |
| UESTC-MMEA-CL Xu et al. (2023) | 2023 | 10 | 32 | D | In/Out | head | Head | Trimmed |
| **WEAR** | **2023** | **18** | **18** | **S** | **Out** | **Head** | **Limbs** | **Untrimmed** |

limits the amount of variety captured in the visual data as lighting conditions and surroundings remain the same throughout all participants. Further, as cooking usually involves object-centric activities, we deem said datasets be more biased towards vision-based prediction scenario, with most of the action taking place in the POV of the user. Compared to Song et al. (2015) and Diete et al. (2019), WEAR provides a larger participant count and, unlike Song et al. (2015) continuous instead of clip-based data streams. Especially the latter ensures that algorithms are assessed in their ability to differentiate unrelated actions (like breaks) from relevant activities, being a necessary trait of HAR prediction algorithm in order to be applied in-the-wild (Bulling et al., 2014).

With early works such that of Spriggs et al. (2009) having shown the complementarity of inertial- and camera-based features, research has followed up by exploring different ways of combining the two modalities. One can categorize such methods broadly by the point in time at which the fusion of both modalities is performed. Late fusion approaches usually follow a two-stream architecture training both vision- and inertial-based modalities separately before merging together outputs of each stream through such as produced softmax probabilities e.g. via a weighted combination (Wei & Kehtarnavaz, 2020), pooling operations (Song et al., 2016a; Imran & Raman, 2020b), majority voting (Diete et al., 2018) or a concurrent classifier (Wu et al., 2017; Diete & Stuckenschmidt, 2019; Ijaz et al., 2022). Early fusion approaches aim at jointly learning from both modalities by using feature embeddings calculated on one (or both) modalities to e.g. use the concatenation of both to train a concurrent network (Imran & Raman, 2020a; Xu et al., 2023; Nakamura et al., 2017; Lu & Velipasalar, 2018; Hu et al., 2023; Ehatisham-Ul-Haq et al., 2019; Diete & Stuckenschmidt, 2019; Diete et al., 2019; Song et al., 2016b; Yu et al., 2019; Chen et al., 2016; Islam & Iqbal, 2022; 2021), enhance softmax probabilities used during late fusion (Diete & Stuckenschmidt, 2019; Diete et al., 2019) or adding intermediate cross-view connections amongst the two modality streams (Ijaz et al., 2022). With experiments showing that single-stage temporal action localization models are able to produce competitive results on raw inertial data, this paper also tests the applicability of two state-of-the-art models, namely the ActionFormer and TriDet model, to fuse and combine cues of both modalities in an early-fusion style. Unlike other early fusion techniques, our approach is the first to directly use the raw inertial data by means of simple concatenation together with a vision-based feature embedding.

## 3 METHODOLOGY

**Study Design & Scalable Pipeline**     Participants were recorded during separate recording sessions. Prior to their first session, participants were handed a recording plan which outlined the study protocol as well informed about any risks of harm, data collection, usage, anonymisation and publication, as well as how to revoke their data usage rights at any point in the future. The study design involving human participants was reviewed and approved by [Anonymized]. All participants were briefed and provided their written informed consent. Each participant was asked to perform 18 workout activities. The location and the time of day at which the sessions were performed, were not fixed and

thus vary across subjects. Participants were suggested to follow a two-session setup, i.e. 9 activities per session. Nevertheless, it was allowed to differ from this setup and split the 18 activities across as many (or as few) sessions as participants liked. This caused the amount of recording sessions to vary across subjects, but also increased the amount of captured variability in weather conditions and recording locations. In order to avoid misunderstandings in the execution of the activities, the authors discussed all activities prior to each session and encouraged participants to ask questions during the session if something remained unclear. Participants were tasked to perform each activity for roughly 90 seconds. As activities varied in their intensity, it was not required to perform activities for 90 seconds straight and participants could include breaks as needed. Furthermore, to ensure that each participant was able to perform all workout activities properly, the recording plan detailed how activities could be altered in their execution, for instance so that they required less physical strength. The recording plan provided with our dataset (see Section E in the supplementary material) includes all necessary materials and is written in such a way that all activities and sessions can easily be reproduced by persons other than the authors. Besides the used sensors for video and acceleration recording, the exercises only require a yoga mat and a chair (or similar items). Sessions can be recorded at any location outside as long as the privacy of the participants as well as pedestrians is ensured. We argue that this facilitates reproducibility, and with a minimal setup ensures that it is possible for others to extend our dataset at a later date.

**Participant Information** We recorded data for 18 participants (10 male, 8 female) at 10 different locations and under varying weather conditions over a stretch of 5 months (October till February), totalling more than 15 hours, with each participant on average contributing roughly 50 minutes of data. The participants were at the time of recording on average 28 years old ($\pm$ 5), 175.4 cm tall ($\pm$ 10.8) and weighed 69.26 kg ($\pm$ 12.43). In order to assess their sports level, participants filled in a post-session questionnaire. The questionnaire contained questions related to vital information (such as body height, weight and age), weekly workout frequency (min. 15 minutes duration) and experience in particular workout activities. On average, participants which took part in the study tend to work out 3.6 times per week ($\pm$ 2.1), already knew 15.06 ($\pm$ 3.75) out of the 18 activities in advance, and regularly conduct 5.5 ($\pm$ 3.74) of the recorded activities as part of their private workouts. Participants reported for their personal workout schedules a wide-range of cardio- (running, hiking, cycling, dancing), strength- (weight lifting, freeletics, rowing), team- (volleyball, basketball, table-tennis) and flexibility-focused (yoga, ballet) exercise types.

**Dataset Collection & Structure** The WEAR dataset provides subject-wise raw and processed acceleration and egocentric-video data (see Figure 1). We focus on 3D accelerometers especially as they cover a substantial amount of commercial fitness devices worn at the wrists and ankles. They furthermore are used in a large set of existing research and datasets focusing on wearable data for activity recognition, and they do not suffer from noise, drift, and other device-specific characteristics. 3D accelerometer data was collected at 50 Hz with a sensitivity of $\pm$ 8g using four open-source Bangle.js smartwatches running a custom, open-source firmware (Van Laerhoven et al., 2022). The watches were placed by the researchers in a fixed orientation on the left and right wrists and ankles of each participant. Egocentric video data was captured using a GoPro Hero 8 action camera, which was mounted using a head strap on each participant's head. The resulting '.mp4'-videos were recorded at 1080p resolution with 60 frames per second and the camera being tilted downwards in a 45 degree angle. A second tripod-mounted camera was placed within the proximity of each participant to facilitate annotation recording the environment in which the workout was performed from a third-person-perspective. For privacy reasons, the second camera's video and all audio captured are not part of the WEAR dataset. During postprocessing, the delta-compressed inertial data, extracted from the watch's memory, was decompressed to '.csv'-format. Inspired by the works of Scholl et al. (2019) and Morshed et al. (2022), we made use of the similarities between inertial sensor and audio data and converted the 3D accelerometer data to '.wav'-files, which allowed to import both modalities into a standard video editing software. By having participants perform synchronization jumps, i.e. jumping 3 times while raising the arms during the jump, at the start and end of each session, peaks in the inertial data were able to be mapped to timestamps in the video stream. Lastly, activity labels, which were added as video subtitles, were exported along with the synchronized video and inertial data streams and appended as an additional column to the inertial data as well as provided as '.json'-format files, following the THUMOS-14 Jiang et al. (2014) formatting-style.

## 4 BENCHMARKS AND BASELINE RESULTS

Though the WEAR dataset provides the possibility for a multitude of HAR use cases, this paper focuses on introducing one sample application scenario per data modality, namely: (1) inertial-based wearable activity recognition, (2) vision-based temporal action localization, as well as, (3) a combined approach using both data modalities as input simultaneously. We chose to use said application scenarios because of their similarities with each other as they both aim to detect a set of activities in an untrimmed sequence of data. Nevertheless, other HAR-specific (e.g. action anticipation and classification) and non-HAR application scenarios (e.g. hand detection, pose estimation or simultaneous localization and mapping (SLAM)) are applicable. During each experiment we employ a three-fold validation split each time using 12 subjects for training while reserving 6 subjects for validation. The validation is applied in such a way that each subject becomes part of the validation set exactly once with the final evaluation metrics being the average across the three splits. In order to minimize the risk of performance differences between experiments being the result of statistical variance, evaluation metrics are averaged across three runs each time employing a different random seed. With the standard error of evaluation metrics amongst runs being at maximum 2.5% and the majority of runs being below 1%, we only report average evaluation metrics in this paper. All mentioned experiments were conducted on a single NVIDIA Tesla V100 GPU and lasted no longer than 24 hours. Though sharing inherent similarities, vision-based action localization algorithms predict a collection of activity segments defined by a start and end time, while, contrarily, inertial-based HAR systems provide labels based on the pre-defined windowed segmentation. Given their difference in prediction output, different evaluation metrics are applied, with mean average precision (mAP) being most prominent metric in vision-based temporal action localization and accuracy/ F1-score being the most prominent metric in inertial-based activity recognition. Therefore, to guarantee comparability amongst application scenarios and architectures, predictions of each algorithm are converted such that both vision- and inertial-based evaluation metrics can be calculated. More specifically, our reported benchmark evaluation metrics are (1) a record-based calculated recall, precision and F1-score, and (2) segment-based mean average precision (mAP) at different temporal intersection over union (tIoU) thresholds, commonly used to evaluate temporal action localization datasets.

**Vision-based Temporal Action Localization** Same as Zhang et al. (2022) and Shi et al. (2023), we chose to train the vision-based benchmark models using two-stream I3D feature embeddings pretrained on Kinetics-400 applying three different clip lengths (0.5, 1 and 2 seconds) with a 50% overlap between clips. Besides increasing the number of epochs to 300, we chose to use the same training strategy which produced best performing results on the EPIC-Kitchens dataset (Damen et al., 2022) as reported by both architectures. Different from inertial-based approaches, temporal action localization models are not trained and able to predict an explicitly modelled NULL-class. With both models being set to predict up to 2000 action segments per video, each timestamp ended up being classified by an action segment causing prediction performance of the NULL-class being (close to) 0% accuracy. We therefore eliminated low-scoring segments by increasing the scoring threshold of both models to be 0.2, which significantly increased the accuracy of the NULL-class, while only marginally affecting prediction performance of all other activity classes (see Section C.3 of the supplementary material for an ablation study). Looking at results presented in Table 2 one can see that for the vision-based models, a clip length of 1 second delivered the best predictive performance. Analysing per-class results, one can see that the vision-based approaches struggle differentiating between different running styles, activities which do not take place within the field of view of the participant (e.g. triceps stretches) as well as normal and complex sit-ups.

**Inertial-based Wearable Activity Recognition** As our inertial-based benchmark algorithms of choice we use the shallow DeepConvLSTM proposed by Bock et al. (2021) and the Attend-and-Discriminate model proposed by Abedin et al. (2021). During all experiments we employed the same training strategy as suggested by Bock et al. (2021), which showed to produce reliable results on a multitude of inertial-based HAR datasets, only increasing the number of epochs to be the same as during the vision-based experiments (see Section C.2 of the supplementary material). To compensate with longer training times, we applied a step-wise learning rate schedule. Further, incorporating architecture changes suggested by Bock et al. (2021), we altered the Attend-and-Discriminate model to use a one-layered instead of a two-layered recurrent module and scaled the convolutional kernel size according to the sliding window and sampling rate of the WEAR dataset

Table 2: Results of human activity recognition approaches based on body-worn IMU (Inertial), vision (Camera) and combined (Inertial + Camera) features for different clip lengths (CL) on our WEAR dataset evaluated in terms of precision (P), recall (R), F1-score and mean average precision (mAP) for different temporal intersection over union (tIoU) thresholds. The results underline the complementarity of the inertial and camera modalities. Best results per modality are in **bold**.

| | Model | CL | P | R | F1 | mAP 0.3 | 0.4 | 0.5 | 0.6 | 0.7 | Avg |
|---|---|---|---|---|---|---|---|---|---|---|---|
| Inertial | Shallow D. | 0.5 | 86.77 | 75.42 | 79.18 | 54.36 | 51.67 | 49.42 | 47.40 | 44.70 | 49.51 |
| | A-and-D | 0.5 | 87.54 | 75.98 | 79.59 | 53.57 | 51.08 | 48.51 | 45.82 | 42.87 | 48.37 |
| | ActionFormer | 0.5 | 78.73 | 70.50 | 72.51 | 63.71 | 61.28 | 53.90 | 39.81 | 26.40 | 49.02 |
| | TriDet | 0.5 | 86.06 | 70.10 | 75.25 | 66.01 | 63.71 | 57.70 | 49.30 | 41.09 | 55.56 |
| | Shallow D. | 1 | 88.02 | 77.03 | 80.86 | 57.09 | 55.32 | 53.61 | 50.59 | 47.85 | 52.89 |
| | A-and-D | 1 | 87.87 | 79.02 | 82.01 | 56.38 | 54.47 | 52.28 | 50.07 | 46.92 | 52.03 |
| | ActionFormer | 1 | 81.69 | 75.37 | 76.86 | 72.90 | 71.30 | 68.28 | 64.14 | 56.65 | 66.65 |
| | TriDet | 1 | 83.85 | 73.76 | 77.12 | **73.27** | **71.66** | **69.83** | **66.79** | **62.25** | **68.76** |
| | Shallow D. | 2 | 87.92 | 78.16 | 81.60 | 59.89 | 57.00 | 54.69 | 51.77 | 48.99 | 54.47 |
| | A-and-D | 2 | **88.24** | **80.55** | **83.08** | 58.32 | 56.68 | 54.44 | 51.58 | 48.34 | 53.87 |
| | ActionFormer | 2 | 78.18 | 69.15 | 71.15 | 66.43 | 63.30 | 60.47 | 56.66 | 50.26 | 59.43 |
| | TriDet | 2 | 81.72 | 69.37 | 72.53 | 65.57 | 63.65 | 61.86 | 59.07 | 54.82 | 60.99 |
| Camera | ActionFormer | 0.5 | 68.06 | 57.68 | 58.47 | 51.27 | 49.45 | 45.74 | 36.10 | 23.38 | 41.19 |
| | TriDet | 0.5 | 73.21 | 57.73 | 60.99 | 53.41 | 51.19 | 47.24 | 40.80 | 35.08 | 45.54 |
| | ActionFormer | 1 | 72.63 | **68.87** | 67.26 | 63.99 | 62.32 | 60.62 | 57.88 | 52.79 | 59.52 |
| | TriDet | 1 | **75.32** | 68.07 | **67.95** | **64.36** | **63.30** | **61.38** | **59.13** | **54.64** | **60.56** |
| | ActionFormer | 2 | 69.67 | 65.79 | 64.15 | 61.32 | 59.92 | 57.96 | 55.91 | 50.39 | 57.10 |
| | TriDet | 2 | 73.85 | 64.09 | 64.25 | 60.95 | 60.03 | 57.75 | 55.55 | 52.19 | 57.30 |
| Inertial + Camera | ActionFormer | 0.5 | 82.40 | 70.96 | 73.76 | 64.95 | 63.89 | 58.49 | 44.67 | 31.77 | 52.75 |
| | TriDet | 0.5 | **87.85** | 70.34 | 75.90 | 67.65 | 66.05 | 62.22 | 55.55 | 46.12 | 59.52 |
| | ActionFormer | 1 | 82.38 | 80.30 | 80.15 | 77.63 | 75.97 | 73.28 | 70.31 | 63.04 | 72.05 |
| | TriDet | 1 | 84.99 | 79.55 | 81.08 | 78.64 | 77.45 | 75.74 | 73.40 | 68.79 | 74.81 |
| | ActionFormer | 2s | 79.19 | 73.88 | 74.52 | 71.10 | 68.79 | 66.38 | 63.00 | 57.54 | 65.36 |
| | TriDet | 2s | 83.10 | 74.55 | 76.72 | 71.20 | 69.69 | 67.88 | 65.49 | 61.77 | 67.20 |
| | *O-LF(I, C)* | *0.5s* | *96.19* | *89.32* | *92.13* | *75.96* | *74.06* | *71.90* | *69.54* | *68.32* | *71.96* |
| | *O-LF(I, C)* | *1s* | *95.52* | *91.52* | *93.08* | *74.86* | *74.09* | *72.78* | *71.68* | *70.23* | *72.73* |
| | *O-LF(I, C)* | *2s* | *94.99* | *91.03* | *92.46* | *73.71* | *72.99* | *71.88* | *70.26* | *68.95* | *71.56* |
| | *O-LF(I, C, I + C)* | *0.5s* | *97.64* | *91.75* | *94.27* | *82.74* | *81.38* | *79.89* | *78.08* | *77.33* | *79.88* |
| | *O-LF(I, C, I + C)* | *1s* | *97.08* | *94.52* | *95.59* | *83.56* | *83.16* | *82.38* | *80.96* | *79.83* | *81.98* |
| | *O-LF(I, C, I + C)* | *2s* | *97.20* | *93.40* | *94.96* | *83.18* | *82.61* | *81.87* | *80.71* | *80.71* | *79.62* |

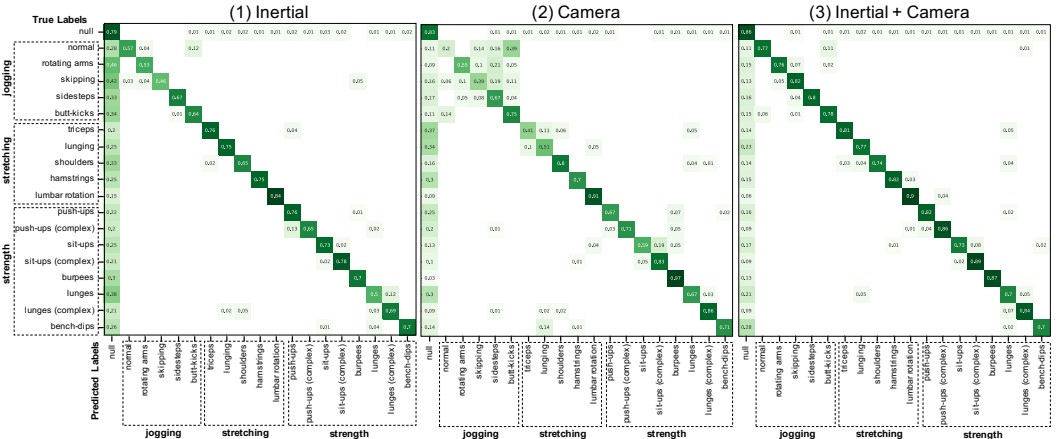

Figure 2: Confusion matrices of the TriDet model (Shi et al., 2023) being applied using inertial, vision (camera) and both combined (inertial + camera) with a one second sliding window and 50% overlap. Compared to inertial-based architectures (Bock et al., 2021; Abedin et al., 2021) overall confusion (except for the NULL-class) is decreased. After combination strengths of each architecture are leveraged with e.g. jogging activities not getting confused anymore and overall confusion with the NULL-class decreases. Note that confusions which are 0 are omitted.

(see Section C.1 of the supplementary material for further details). As the inertial-based architectures are providing predictions on a per-window basis, intermediate, short-lasting activity switches occur quite frequently along the time axis causing said architectures to produce only small coherent segments and ultimately lower mAP scores compared to the vision-based models presented in this paper. In order to remove these intermediate switches, predictions made by the inertial-based architectures were smoothed using a majority-vote-filter of 15 seconds (see Section C.3 of the supplementary material for an ablation study on the performed postprocessing). With the confusion of vision-based models being mostly among the activity categories (jogging, stretching and strength), inertial-based models show a larger degree of overall confusion among all workout classes. Caused by per-window predictions and resulting intermediate activity switches, calculated mAP scores of the inertial-architectures are significantly lower than that of the camera-based approaches. Nevertheless, one can see that inertial-based models are on average able to predict all workout activities more consistently and produce the highest classification metrics across all experiments.

**Multimodal (Inertial and Egocentric Video) HAR** Within our last set of experiments, we assess the applicability of single-stage temporal action localization models for inertial-based as well as modality-combined (inertial + camera) HAR. In order to early fuse the two-stream I3D feature embeddings with the inertial data, we flattened the windowed inertial data such that the captured acceleration along each axis of each sensor is appended to become a vector of size [*window length × no. sensor axis*]. Using the same hyperparameters as used during vision-based experiments, a plain ActionFormer and TriDet network are not only able to be produce competitive classification results based on inertial input data, but, unlike the inertial-based architectures, show less confusion amongst the activity classes. Furthermore, with both temporal action localization models predicting segments instead per-window activity labels, mAP scores significantly increase. By means of simple concatenation of both modalities, both architecture achieve the highest average mAP and close-to-best F1-scores across all experiments (see Table 2). Comparing confusion matrices of all three approaches (see Figure 2) reveals that both vision models, applied in a plain fashion, are able to successfully combine inertial and vision data and leveraging the strengths of each modality.

To assess how our earl-fusion approach compares to voting-based late-fusion approaches such as proposed by Ijaz et al. (2022), we implemented an *Oracle*-based late fusion, which creates perfectly late fused predictions of different models. The predictions are merged by comparing each of them with the ground truth data and only keeping, if predicted by one of the networks, the correct prediction. Interestingly, the first *Oracle*-late-fusion *O-LF(I, C)*, which late fuses predictions of the best inertial and best vision model, produces lower mAP scores than that of the best temporal action localization model being trained on both modalities simultaneously. Furthermore, late-fusing the best inertial, vision and early-fusion approach (*O(I, C, I + C)*), increases mAP scores of *O(I, C)* by as much as 10%, suggesting the early-fusion-based approach is capable of learning to differentiate activities both single-modality models failed to classify correctly. Nevertheless, classification results of the *Oracle*-based late fusion significantly outperform both single- and combined-modality approaches, indicating that the data set is far from being saturated.

## 5 LIMITATIONS

Our dataset contributes a benchmark for human activity recognition classifiers, for the two leading wearable modalities of egocentric video and inertial data, using in particular a high variety of fitness exercises and outdoor scenes. With the current selection of participants, the WEAR dataset is biased towards young, healthy people. Given the ease of reproducibility, future extensions of the WEAR dataset could focus on featuring participants (1) of an older and/ or younger age, (2) with known physical impairments and (3) sessions recorded at new locations (outside of [Anonymised]) and at different times of the year (e.g. summer). As supplementary experiments already indicate (see Section C.7 in the supplementary material), recording the same participants a second time would allow to analyse how a certain degree of familiarity with the recording setup can be seen in altered movements (e.g., via a smoother execution of activities) as well as give an intuition about robustness of learned approaches. Besides extending the amount of data recorded, further recordings could also involve other sensors such as higher-end commercial smartwatches to enable the study of increased sampling rates, the variability of the capturing devices, and the inclusion of additional

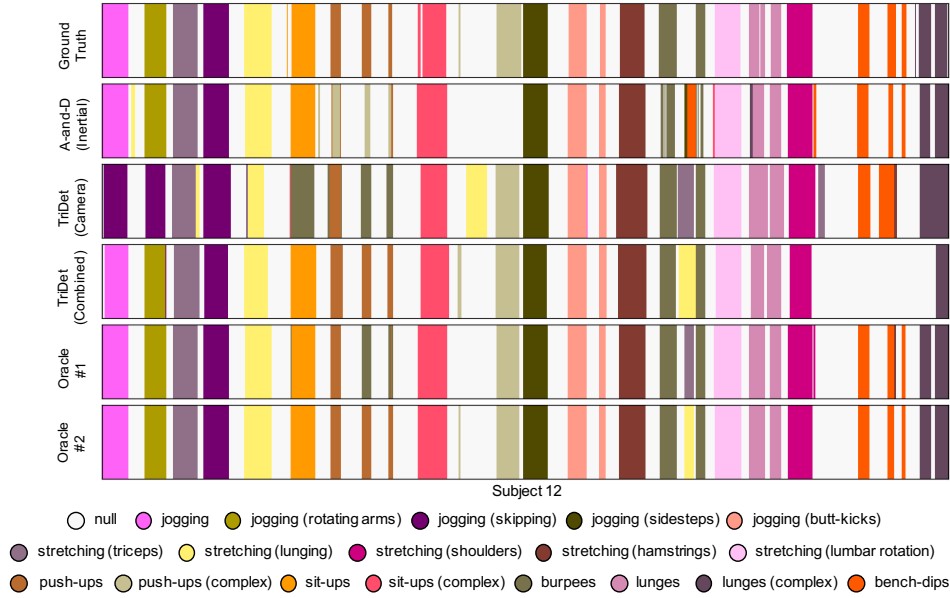

Figure 3: Color-coded comparison of the ground truth data of a sample subject with the best inertial-based (A-and-D), camera-based (TriDet) and fusion-based model (TriDet) along with an oracle combination of the best fusion-based model (*O(I, C)*) as well as an oracle combination the best camera, inertial and fusion-based-model (*O(I, C, I + C)*) using a sliding window approach of 1.0 seconds with a 50% overlap. The visualisation underlines the similarities amongst the predictive streams of *O(I, C)* and the fusion-approach as well advantages of learning from both modalities simultaneously.

modalities such as 3D gyroscopes, 3D magnetometers, or photoplethysmography (PPG) to obtain fitness-relevant information such as heart rate), as well as additional wearables, such as earables.

## 6    CONCLUSION

In this paper, we introduced a benchmark dataset for both inertial- and vision-based Human Actvity Recognition (HAR), to explore the learning of HAR across these modalities. The dataset comprises data from 18 participants performing each 18 different sports activities with the two common types of wearable sensors delivering inertial (3D acceleration) and camera (egocentric video) data. Our WEAR dataset provides a challenging prediction scenario across both modalities marked by purposely introduced activity variations along with a small information overlap between the inertial and vision data, putting forward the necessity of exploring techniques to combine both modalities.

Benchmark results obtained using each modality separately show that each modality interestingly offers complementary strengths and weaknesses in their prediction performance. In light of the recent successes of temporal action localization following the architecture design as proposed by Zhang et al. (2022), we demonstrate their versatility by applying them in a plain fashion using only inertial data as input. Results show that the vision-based models are not only able to produce competitive results using inertial data, but also can function as an architecture to fuse both modalities by means of simple concatenation with vision data. In experiments that combined raw inertial with extracted vision-based feature embeddings, the plain, vision-based temporal action localization models were able to produce the highest average mAP and close-to-best F1-scores. Lastly, to give an intuition about a possible upper bound for future fusion-approaches, we evaluated an oracle-merged late fusion of the best inertial- and vision-based model predictions.

Vision-based temporal action localization such as the ActionFormer (Zhang et al., 2022) have thus far neither been explored in inertial nor in the combination of inertial- and vision-based human activity recognition. With WEAR, we provide both communities (inertial- and vision-based HAR) a common, challenging benchmark dataset to assess the applicability of combined approaches.

# 7    ETHICS STATEMENT

Before participating in the study, participants were notified that by nature the data they provide can only be pseudonymised. This means that, though requiring a substantial amount of effort, the identity of a person can be reconstructed. Although participants agreed to include their egocentric videos in a public dataset, it is essential to refrain from actively identifying the individuals featured in the WEAR dataset. If other researchers decide to contribute to the WEAR dataset by recording additional participants, societal and ethical implications should be considered. As with the participants part of the original release of the WEAR dataset, all participants must be briefed before their first recording, making them aware of all necessary information and implications that come with providing to the WEAR dataset. Recording locations should only be chosen if video recordings are allowed at said location and participants are given enough space to perform each activity safely. If the recording location involves pedestrians walking within close proximity, pedestrians should be notified that they are being recorded and, if applicable, captured faces should be blurred during postprocessing.

The WEAR dataset and associated code are made public for research purposes. With the accurate detection of physical activities that we perform in our daily lives having been identified as valuable information, the WEAR dataset focuses on one of the most popular application scenarios of wearable smartwatches and action cameras, i.e. self-tracking of workout activities. With the ease of reproducability we hope to make WEAR a collaborative, expanding dataset which researchers from different locations and backgrounds can contribute to. For example, as the current selection of participants is biased towards healthy, young people, we hope to overcome said limitation by including people from more diverse backgrounds and age groups in future iterations of the dataset.

Lastly, the authors took great care of avoiding any infringement of rights during the data collection process. Yet, in case of conflicts, they are of course committed to taking appropriate actions, such as promptly removing data associated with such concerns.

# 8    REPRODUCIBILITY STATEMENT

The source code that was used to conduct all experiments is available via [Anonymized] (`https://www.anonymous.edu/anon`). A snapshot of the code is provided as part of the supplementary material download. The repository is written in such a way that other architectures (both inertial- and vision-based) can be added in the future. The repository provides Readme files which give details on the overall structure of the repository, how collect additional data and how to set up an Anaconda environment with the needed packages to run experiments. Experiments are defined via '.json'-format configuration files which allow for easy sharing of used hyperparameter settings.

WEAR and all associated files are offered under a Creative Commons Attribution-NonCommercial-ShareAlike 4.0 International License. The dataset is hosted via a cloud-storage platform. It is a non-commercial cloud storage service for research, studying and teaching and is provided to participating institutions exclusively. With locations exclusively in [anonymized], [anonymized] is subject to the strict directives on data protection and data security. The dataset download is structured into the (1) '.json'-formatted annotations, (2) raw, synchronized inertial and vision data and (3) precomputed feature embeddings as mentioned in the main paper. Third party data-hosting services will be explored once the dataset paper is published and in a non-changing state. We will involve the ethics council of [anonymized] during our decision process to ensure a each selected hosting platform is inline with our data privacy standards. Note that to ensure anonymization of affiliated authors, the dataset cannot be shared as part of the review phase, making it not possible to rerun experiments.

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
