# OpenReview forum: "WEAR: An Outdoor Sports Dataset for Wearable and Egocentric Activity Recognition"
_ICLR.cc/2024/Conference — ICLR 2024 Conference Withdrawn Submission_

### Official Review · Reviewer_Ftjo · 2023-10-24

**Soundness:** 3 good
**Presentation:** 4 excellent
**Contribution:** 3 good
**Rating:** 5
**Confidence:** 3

**Summary:**

This paper introduced a new outdoor sports dataset, namely WEAR, containing egocentric video and sensor data for human activity recognition (HAR), corresponding to vision and inertial-based tasks. There are 18 participants involved engaging in 18 different workout activities at 10 different locations. This dataset provides the potential to research the effectiveness of vision, inertial, and combined inputs on HAR. Besides, the authors also did tons of experiments as benchmark to show the strengths of each locality.

**Strengths:**

1. The tables, figures, and writing in the paper are clear and understandable.
2. The authors did a large number of experiments to show the features of each modality, i.e. vision and inertial, in HAR. Specifically, the vision-based temporal action localization, and Inertial-based wearable activity recognition, and multimodal HAR.

**Weaknesses:**

1. Compared to the vision-based action recognition datasets, the scale of WEAR is relatively small, with only 18 people operating 18 workout activities, and the total duration of each activity is 90 seconds, which does not show the minimum times the activity appears in this period.
2. One of the features of this dataset is that it is recorded in various conditions, such as location, and weather, with limited information overlapping. This proposed data collection method makes it really challenging, due to its variety and small number. This may make the model hard to extract enough meaningful information from this dataset.
3. The participants are young people, which may lead to bias. And there are multiple similar activities among these 18.

**Questions:**

The data collection used a head-mounted camera and smartwatches on the legs and arms. What is the motivation to collect data under these settings? Especially, there are few wearable products on legs in outdoor scenarios.

---

### Official Review · Reviewer_PuSp · 2023-10-30

**Soundness:** 2 fair
**Presentation:** 3 good
**Contribution:** 2 fair
**Rating:** 5
**Confidence:** 3

**Summary:**

This paper introduces WEAR dataset, an outdoor sports dataset for both vision- and inertial-based human activity recognition.

The dataset contains data from 18 participants performing a total of 18 different workout activities with untrimmed inertial (acceleration) and camera (egocentric video) data recorded at 10 different outside locations.

WEAR dataset provides a challenging prediction scenario marked by purposely introduced activity variations as well as an overall small information overlap across modalities.

**Strengths:**

+ The concepts are clearly introduced and explained, nice motivation.

+ Nice and interesting comparisons presented in Table 2.

+ Nice visualisations presented in the experimental section.

**Weaknesses:**

Major:

- The contributions highlighted in the Introduction section are a bit vague. What are the major contributions w.r.t. dataset collection, benchmarks and baselines etc?

- The collected dataset seems still quite limited w.r.t. the number of classes and the number of performing subjects, which is a noteworthy shortcoming. The major differences compared to existing datasets are not clearly highlighted, or only limited discussions provided. More insights in terms of different aspects/perspectives/views should be added and discussed.

- The experiments w.r.t. benchmarks and baseline results are very limited w.r.t. different scenarios/tasks. This is also a noteworthy shortcoming.

- The evaluation metrics are also very limited. What are the other scenarios that this dataset can be used for? It is suggested to dig deeper into the problem rather than just propose a dataset and simply setup benchmarks and baselines.

- I noticed that the authors have a separate section talking about the limitations of the work; however, this section is not very well-written and it is suggested to improve it. For each limitation/weakness, it is suggested to provide detailed plans and solutions etc.

Minor:

- Page 1 spelling mistakes, e.g., timeseries ->  time series

- Fig. 1 some fonts are too small to read.

- Left quotation marks require correction e.g., page 5 ‘dataset collection & structure’ section.

**Questions:**

Please refer to weakness section

**Details Of Ethics Concerns:**

The authors provided some information in the paper.

---

### Official Review · Reviewer_1mvR · 2023-10-31

**Soundness:** 3 good
**Presentation:** 3 good
**Contribution:** 2 fair
**Rating:** 6
**Confidence:** 3

**Summary:**

The manuscript introduces a new dataset called WEAR. This dataset contains ego-centric head-worn video data in combination with IMU data from each limb. The dataset is recorded by 18 participants. Each participant recorded 18 different activities. The data contains a total of 15h of this multi-modal data.

**Strengths:**

The dataset seems constructed well for a novel set of tasks in the fitness space.

The recording of IMU signals from all limbs allows interesting experiments and ablations of which limb or modality yields the most signal on which activity.

There is a wealth of baseline experiments of state of the art models that can use IMU or vision or both imu and vision data. This sets useful baseline performance on the dataset.

The Oracle experiments convincingly show that there is quite a gap from the current performance to an "optimal" combination of the two modalities. This clearly shows there is more research to be done (even if the dataset at hand may be to small to actually help train models to close that gap).

The manuscript is well written and illustrated except for a few details (see weaknesses and questions).

**Weaknesses:**

The primary weakness of the dataset for ML is its relative small size of only 15h. (Compare that to 3670h of data in Ego4d). This makes it not well suited for training generalizable models solely on it (as evidenced by the need to pretrain on Kinetics 400). It may still be able to serve as a validation dataset for how well modalities can be combined on for fitness activity recognition.

Temporal alignment is critical for this kind of multi-sensor multi-modal data. The use of 4 jumps to get alignment signal is a bit questionable for alignment, given that IMU as running at 50Hz and camera at 60Hz. It will depend a lot on the implementation how good this alignment is. Unfortunately this details is not provided.

From Fig 2 it seems like the IMU-based models have low confusion, whereas the camera-based ones have more. Combining them does not seem to change much with confusion in general except for helping with the null-class. It kind of raises the question what vision is good for?

The implicit assumption in the evaluation and dataset collection as presented is that in the future data will be available from both wrists and both feet which seems doubt full. Another interesting experiment would be to ablate which limb location makes the biggest difference in the action localization task. It would also be interesting to see the differences between confusion matrices between the different limbs. If only IMU data from one limb is used maybe the camera will also show a larger impact on performance.

**Questions:**

- whats cls in table 1?
- whats the number of hours collected for each dataset in table 1?
- It is a bit unclear for the different models how they were trained on the relatively small dataset (12 participants data is only 10h of data which is very small for Transformer-based models) or whether they were pretrained on a different dataset before finetuning on the WEAR dataset. It does sound like only the vision based models were pretrained on the Kinetics 400 dataset. Otherwise all models were trained only on the WEAR dataset?

---

### Official Review · Reviewer_55jH · 2023-11-05

**Soundness:** 3 good
**Presentation:** 3 good
**Contribution:** 2 fair
**Rating:** 3
**Confidence:** 4

**Summary:**

The paper introduce a new benchmark dataset, called WEAR, for human activity recognition using both inertial (acceleration) and camera (egocentric video) data. The dataset includes data recorded by 19 participants performing 18 different sports activities at 10 outdoor locations. The paper presents three benchmark tasks : inertial-based activity recognition, vision-based temporal action localization, and multimodal (inertial + video) activity recognition. Experiments show that recent action localization models (e.g., ActionFormer) can be extended for multimodal activity recognition and achieves superior results compared with single-modality or early fusion based methods.

**Strengths:**

The paper introduces a new dataset for outdoor physical activity recognition with multimodal (e.g., video and inertial) data, which is limited in the existing work. The dataset can be used for developing and evaluating inertial based or multimodal based models for physical activity recognition.

**Weaknesses:**

The scale of this dataset is too small for sufficient training of recent models and evaluation of model generalization. The dataset only involves 18 subjects (12 for training and 6 for testing), with only 15 hours in total (untrimmed) and roughly 8 hours for foreground activities. This data scale is too small to train a video model especially considering the increasing scale of the recent models, and it also makes the evaluation less convincing. The dataset is also less valuable for developing strong, generalizable multimodal activity recognition models due to the limited variation (only 18 activities with fine-grained differences, 10 locations and 18 subjects).

**Questions:**

N/A

---

### Author Response · Authors · 2023-11-21

Dear reviewers,

Thank you for the valuable feedback and constructive comments on our submission. In the following we want to address raised concerns from your side.

**Limited Size and Variety of Dataset.** The WEAR dataset is most comparable to the CMU-MMAC, ActionSense, ADL and MEAD dataset. Crucially, other datasets mentioned in Table 1 that include the Ego4D and EPIC-Kitchens dataset are not providing wrist-worn IMU data. With commercial fitness trackers readily available, we deem inertial data captured at the wrists to be most descriptive in the context of sports activity recognition. Among these four datasets being most similar to WEAR, our dataset provides the largest participant count while being comparable in size and activity count to the ActionSense dataset. Further, we argue that we do provide a challenging prediction scenario, as all activities were recorded outside with most activities not depending on nearby objects in the users' field of view. Lastly, each activity was executed by each participant for at least 90 seconds (excluding breaks).

**Synchronisation Process.** In order to properly synchronize the inertial and video data we converted the inertial sensor streams to ‘.wav’ files, such that we could import them to Final Cut and see the acceleration as a graph-like representation. This allowed a frame-by-frame synchronization of the complete stream, correcting the initial synchronization based on the jumps performed at the start and end of each workout. These details can be found in section B.3 of the supplementary material.

**Strengths of each modality.** Related to our dataset research such as that of Spriggs et al. (2009) have shown that inertial and camera data truly have complementary strengths. Though achieving higher classification scores in recognizing the sports activities, approaches trained using only inertial data struggle to differentiate the activities from the NULL-class. The reliable detection of the NULL-class represents one of the key challenges to overcome for HAR being applied on untrimmed sequences. A higher NULL-class accuracy of the camera-based approaches is thus to be considered of high significance. Further, when combining both modalities, we achieve the highest NULL-class accuracy while preserving the recognition performance on the sports activities, validating the effectiveness of the fusion approach.

**Motivation behind dataset and sensor selection.** Our dataset focuses on fitness activity detection by combining  the two leading modalities that have been separately most researched in this area. We added the ankle-based locations to reflect a set of running- and fitness devices known as anklets (or smart/bluetooth socks), and collect inertial data for both wrists and ankles to accommodate for left- and right-handed users. The application scenario of fitness activities has led to the choice of sensors and their acquisition frequency. An ablation study on the importance of each limb location can be found in section C.6 of the supplementary material.

**Details on the training pipeline.** All models part of the benchmark analysis were not pretrained and trained using only each split’s data of 12 participants. Feature embeddings used to train the temporal action localization models on camera data were extracted I3D features, with the feature extractor being pretrained on the Kinetics-400 dataset.

We hope that the above explanations address your concerns and thank you for your comments. Yet, at the current point, we decided to withdraw our manuscript for a resubmission in the future.

Sincerely, the authors